# Age Is a Determining Factor of Dry Eye-Related Signs and Symptoms

**DOI:** 10.3390/diagnostics10040193

**Published:** 2020-03-31

**Authors:** Masahiko Ayaki, Kazuno Negishi, Motoko Kawashima, Miki Uchino, Minako Kaido, Kazuo Tsubota

**Affiliations:** 1Otake Clinic Moon View Eye Center, Yamato 242001, Japan; 2Department of Ophthalmology, Keio University School of Medicine, Tokyo 1608582, Japan; motoko326@gmail.com (M.K.); uchinomiki@yahoo.co.jp (M.U.); fwiw1193@mb.infoweb.ne.jp (M.K.); tsubota@z3.keio.jp (K.T.)

**Keywords:** aqueous deficient dry eye, risk factor, dry eye, evaporative dry eye, reflex compensation

## Abstract

Purpose: The reported signs and symptoms of dry eye (DE) have been discordant. This study evaluated risk factors of DE-related symptoms and signs to explore their association with patient demographics, focusing on the age factor. Methods: The study enrolled 704 consecutive patients visiting general eye clinics who complained of ocular discomfort, but had normal vision. The patients were asked about the presence of six common symptoms related to DE and, tear break-up time (TBUT). The severity of patients’ keratopathy was also examined, and patients underwent Schirmer’s test. Results: Logistic regression analysis demonstrated that younger age (≤29 years) was associated with non-visual symptoms and keratopathy, while older age (≥60 years) was associated with short TBUT and low values on Schirmer’s test. Middle age was associated with both severe symptoms and signs. Conclusions: Discrepancies in the signs and symptoms of DE may depend, in part, on age, with younger subjects showing severe non-visual symptoms with apparently normal tear function and severe keratopathy, and older subjects showing fewer symptoms and less severe keratopathy despite worse tear function.

## 1. Introduction

Dry eye (DE) is a complex disease and its clinical manifestations vary on a case-by-case basis. According to the latest criteria and definitions [1,2], a diagnosis of DE is made on the basis of corneal and conjunctival epitheliopathy, Meibomian gland dysfunction (MGD), aqueous tear secretion, tear osmolarity, and a variety of subjective symptoms related to DE. However, clinical presentation is often complicated because some patients have severe symptoms without notable corneal findings, and others have no symptoms even though they may present with severe keratopathy and a short tear break-up time (TBUT). Although many investigators have described a disassociation between the signs and symptoms in DE patients [3,4], we assume the presence of corneal hypoesthesia may be associated with dryness and other symptoms [5]. Changes in corneal sensitivity may depend on corneal neurodegeneration and aging [6,7,8]. Other factors that could determine the severity of DE include age, sex, and various environmental and individual factors [9,10,11,12].

Bron et al. [13] proposed that the condition differed between two types of DE, namely aqueous deficient DE (ADDE) and evaporative DE (EDE). The authors suggested that in the initial stages of EDE with MGD and severe symptoms, patients may experience hypersecretion of tears as a compensatory mechanism. As a result of inflammation, thinning of the tear film and progression of lacrimal damage can occur, increasing tear osmolality and damaging the corneal nerve. Consequently, there is a decrease in both compensatory secretion and symptoms. In ADDE, lacrimal failure induces tear hyperosmolality, followed by inflammation and nerve damage. There is also a hybrid form of DE. This proposal provides a reasonable explanation for the variety of signs and symptoms seen in patients with DE.

In this study, we attempted to address the possible limitations of epidemiological studies. Firstly, some questionnaire-based surveys of community-dwelling subjects lack objective diagnostic testing: in the present study, one of the authors (MA, an ophthalmologist) examined and interviewed all the patients face to face. Secondly, most studies have focused on older populations, with few studies including younger participants [14,15,16,17]. We were able to recruit many young patients who presented with a complaint of ocular discomfort to the general eye clinics that participated in this study. Thirdly, many epidemiological studies, such as the Dry Eye-Related Quality-of-Life Score (DEQs) [18], have used self-administered questionnaires to assess symptomatology, which is expressed as a composite score, yielding a combination score for visual, non-visual, psychological, disability, and social issues. In the present study, we investigated six representative symptoms to evaluate the association between each symptom and corneal parameters, including keratopathy, TBUT, and aqueous tear secretion. Fourthly, we examined many treatment-naïve patients attending the clinics on a walk-in basis, so that the natural history of DE could be observed.

The aim of this hospital-based study was to explore associations between demographics and DE-related signs and symptoms across all generations in Japan, based on corneal evaluations and routine ophthalmological examinations. The study was performed over a 3-year period to address seasonal variations in DE status, as reported previously [12].

## 2. Materials and Methods

### 2.1. Study Participants and Institutional Review Board Approval

The present cross-sectional case-control study was conducted in the following settings: Komoro Kosei General Hospital (Nagano, Japan); Shinseikai Toyama Hospital (Toyama, Japan); Tsukuba Central Hospital (Ibaraki, Japan); Jiyugaoka Ekimae Eye Clinic (Tokyo, Japan); Todoroki Eye Clinic (Tokyo, Japan); and Takahashi-Hisashi Eye Clinic (Akita, Japan). The institutional review boards and ethics committees of Keio University School of Medicine, Komoro Kosei General Hospital, Tsukuba Central Hospital, and Jiyugaoka Ekimae Eye Clinic approved this study, and the study was performed in accordance with the Declaration of Helsinki. The other eye clinics participating in the study were described as collaborators in the ethics committee document, and were thus covered under the approval granted by the Keio University School of Medicine, Komoro Kosei General Hospital, Tsukuba Central Hospital, and Jiyugaoka Ekimae Eye Clinic. Informed consent was obtained from all participants.

### 2.2. Inclusion and Exclusion Criteria

The study recruited consecutive patients from participating clinics and hospitals between April 2015 and March 2018. The inclusion criteria were consecutive outpatients with best-corrected visual acuity better than 20/30 in both eyes. The exclusion criteria were any ocular surgery within one month, and any acute eye disease within one week. Consequently, the final study cohort predominantly comprised individuals visiting their clinic for a regular vision check, floaters, or a mild unidentified ocular symptom.

### 2.3. Ophthalmological Examinations

The ophthalmological examination was carried out according to work by Yokoi et al. [19]. The TBUT was measured using wet fluorescein filter paper (Ayumi Pharmaceutical, Tokyo, Japan), applied at the lower lid margin. The TBUT was defined as the time interval between the third blink and the appearance of the first dark spot in the cornea, and was measured using a stopwatch. This was calculated with the mean of three measurements. A corneal staining score was determined to grade corneal epitheliopathy; graded at 0–2 for severity and area. Schirmer’s test was performed without topical anaesthesia. Strips of filter paper (Whatman No. 41; Showa Yakuhin Kako, Tokyo, Japan) were placed for 5 min at the outer third of the temporal lower conjunctival fornix, with the subject blinking spontaneously. The strips were then removed, and the length of the filter paper wetted by the spontaneous blinks was recorded (mm).

### 2.4. Patient Interviews for DE-Related Symptoms

The patients were asked questions to determine the presence or absence of six common DE-related symptoms:dryness; irritation; pain; eye fatigue; blurring; and photophobia. These questions were selected from items on the Dry Eye-Related Quality-of-Life Score (DEQS) questionnaire [18], and were based on the six most prevalent symptoms of DE in patients who had visited the Dry Eye Clinic in the Department of Ophthalmology at Keio University Hospital in 2014. Eye fatigue, blurring, and photophobia were classified as visual symptoms, whereas dryness, irritation, and pain were classified as non-visual symptoms.

### 2.5. Statistical Analysis

Age was stratified by 10-year intervals between the ages of 30 and 69 years, as well as ≤29 and ≥70 years. The prevalence of DE-related symptoms and signs was calculated according to sex. Schirmer’s test, corneal staining score, and TBUT test results from the more severe eye were used for analysis. Using a logistic regression model, odds ratios and 95% confidence intervals were calculated for DE-related symptoms and signs according to age and sex. Data are presented as the mean ± SD or as percentages where appropriate. All analyses were performed using StatFlex (Atech, Osaka, Japan), with *p* < 0.05 considered significant.

## 3. Results

In all, 704 outpatients (mean (±SD) age 54.3 ± 18.0 years; range 8–93 years) participated in this study. The patient demographics are given in (Table 1). The prescribed eyedrops for the treatment of DE were hyaluronate (0.1% sodium hyaluronate), mucin secretagogue (3% diquafosol sodium and 2% rebamipide), and steroid (0.02%/0.1% fluorometholone). Mucin secretagogue was significantly more often prescribed for the older group (*p* = 0.041, Mann–Whitney U Test).

The prevalence of DE-related symptoms and signs in this cohort is given in Table 2. The prevalence of three DE-related non-visual symptoms (dryness, irritation, and pain) was higher in women than in men, and female sex was associated with these symptoms. In contrast, there were no sex differences in the prevalence of visual symptoms (eye fatigue, blurring, and photophobia) (adjusted for age). The results of logistic regression analysis revealed the factors associated with different DE-related symptoms and signs (Table 2 and Table 3). A younger age was associated with keratopathy and the presence of non-visual symptoms. Older age was associated with tear function, including a short TBUT and low Schirmer test value. The female sex was more at risk of non-visual symptoms, short TBUT, and keratopathy, but not low Schirmer test value.

To summarize the findings regarding the effects of age on DE signs and symptoms (Table 4), young subjects had keratopathy with less tear dysfunction, older subjects had less keratopathy and fewer symptoms with severe tear dysfunction, middle-aged subjects had both severe symptoms and signs, and there may be hybrid cases.

## 4. Discussion

The results of the present study indicate that age is closely associated with DE-related symptoms and signs, which is consistent with the natural history of DE, proposed by Bron et al. [13]: specifically, EDE may be predominant in younger patients, the hybrid form of DE may be predominant in middle-aged patients, and ADE may be predominant in older patients. Inflammation plays a major role in DE [20,21,22], and we speculate that many younger patients may be in the acute phase with severe symptoms and keratopathy, whereas most older patients may be in the chronic inflammatory or degenerative phase, with lacrimal damage. In contrast, middle-aged patients tend to occupy a transitional phase, resulting in a mixed clinical presentation depending on corneal sensitivity, tear function, medications, and systemic comorbidities. Middle-aged patients may have non-visual symptoms if their corneal sensitivity is still normal but tear function has deteriorated. Some middle-aged patients may have fewer non-visual symptoms than younger patients, with a short TBUT and low Schirmer’s test values, if their corneal sensitivity is decreased, and some middle-aged patients may be hybrid cases of EDE and ADDE until distinct lacrimal failure and typical ADDE develops. Across all age groups various factors may affect the signs and symptoms of DE [10,11,12,13], and this may be why the clinical manifestation of DE in older patients is sometimes complicated, and a reason for the general inconsistencies in the signs and symptoms of DE.

Based on the present results, different treatments may be applicable for different patient age groups. Anti-inflammatory eyedrops may be considered for younger patients suffering pain, irritation, and dryness. Mucin secretagogue should be first considered for older patients with tear dysfunction. Many middle-aged patients may have complications such as tear dysfunction and other troubling symptoms; in these cases, mucin secretagogue and steroids may be prescribed for clinically significant signs and symptoms.

Short TBUT, severe keratopathy, and non-visual symptoms were more prevalent in women, as was repeatedly reported. Women claim more ocular and physical symptoms than men [14,23,24,25,26,27,28,29]. A previous study [23] of 1518 women with a mean age of 71 years and 581 men with a mean age of 76 years indicated gender differences in DE regarding impact, management, and patient satisfaction. The authors concluded that DE was generally experienced as being more severe among women, as well as having a greater impact on their self-assessed wellbeing. Regarding corneal signs, a study of Japanese office workers [24] indicated shorter TBUT and more severe keratopathy in women. An adolescent study also demonstrated girls had more keratopathy, despite a normal Schirmer’s test value [14]. However, the reasons for these gender differences are still elusive.

The present study has some limitations. Corneal sensitivity, tear osmolality, inflammation markers, and MGD should be examined further as they are all closely associated with symptoms. In addition, the present study is a hospital-based study and not performed on the general population. This limitation is somewhat addressed by the t recruitment of patients attending general eye clinics, where many preclinical DE patients of all generations with either no or mild symptoms were examined by ophthalmologists, as would be the case in a population-based study. Finally, the sexes are not equally represented. The participating eye clinics are walk-in-based, and we recruited patients continuously during the study period. More women visited these clinics, agreed to partake in Schirmer’s test and other tests, and entered the study after inclusion and exclusion criteria than men. Hence, the conclusions on association between gender and DE-related signs and symptoms could be biased by this disproportion. However, the fact that more women visited the participating eye clinic than men may also be a sign of the more frequent occurrence of symptoms among women.

## Figures and Tables

**Table 1 diagnostics-10-00193-t001:** Age and sex distribution and dry eye medication of study participants.

Age (Years)	Sex		Dry Eye Medication		
	Men (*n* = 159)	Women (*n* = 545)	Hyaluronate	Mucin Secretagogue	Steroid
≤29	16 (10.1)	51 (9.4)	22 (32.8)	14 (20.9)	9 (13.4)
30–39	24 (15.1)	58 (10.6)	16 (19.5)	23 (28.0)	5 (6.1)
40–49	31 (19.5)	76 (13.9)	30 (28.0)	24 (22.4)	9 (8.4)
50–59	33 (20.8)	114 (20.9)	44 (29.9)	49 (33.3)	13 (8.8)
60–69	25 (15.7)	114 (20.9)	39 (28.1)	49 (35.3)	13 (9.4)
≥70	30 (18.8)	132 (24.3)	47 (29.0)	51 (31.5)	12 (7.4)
*p* value *	0.019		0.731	0.041	0.543

Data are presented as *n* (%). * Mann–Whitney U Test.

**Table 2 diagnostics-10-00193-t002:** Association between demographics, corneal parameters, and dry-eye-related symptoms.

	Dryness		Irritation		Pain		Eye Fatigue		Blurring		Photophobia	
	Prevalence (%)	OR (95% CI)	Prevalence (%)	OR (95% CI)	Prevalence (%)	OR (95% CI)	Prevalence (%)	OR (95% CI)	Prevalence (%)	OR (95% CI)	Prevalence (%)	OR (95% CI)
Age (years)												
≤29	36/67 (53.7)	1	26/67 (38.8)	1	20/67 (29.8)	1	30/67 (44.8)	1	16/67 (23.9)	1	21/67 (31.3)	1
30–39	45/82 (54.2)	1.04 (0.54–2.00)	22/82 (26.8)	0.57 (0.28–1.14)	14/82 (17.1)	0.51 (0.23–1.12)	46/82 (56.1)	1.51 (0.78–2.90)	26/82 (31.7)	1.48 (0.71–3.09)	23/82 (28.0)	0.89 (0.43–1.82)
40–49	48/107 (44.9)	0.69 (0.3–1.29)	31/107 (29.2)	0.64 (0.33–1.22)	17/107 (16.0)	0.47 (0.22–1.00)	54/107 (50.9)	1.21 (0.65–2.26)	35/107 (33.0)	1.53 (0.76–3.06)	28/107 (26.4)	0.80 (0.40–1.59)
50–59	59/147 (40.1)	0.55 * (0.3–0.99)	32/147 (21.8)	0.42 * (0.22–0.80)	20/147 (13.6)	0.38 * (0.19–0.79)	87/147 (59.2)	1.74 (0.97–3.13)	44/147 (29.9)	1.35 (0.69–2.65)	37/147 (25.2)	0.77 (0.40–1.48)
60–69	52/139 (37.4)	0.50 * (0.27–0.90)	49/139 (35.2)	0.81 (0.44–1.49)	20/139 (14.4)	0.41 * (0.20–0.84)	59/139 (42.2)	0.91 (0.5–1.65)	43/139 (30.9)	1.43 (0.73–2.80)	33/139 (23.7)	0.74 (0.38–1.43)
≥70	35/162 (21.7)	0.22 * (0.11–0.41)	47/162 (29.7)	0.63 (0.34–1.16)	22/162 (13.8)	0.37 (0.18–0.76)	71/162 (44.1)	0.96 (0.53–1.71)	50/162 (31.6)	1.45 (0.75–2.80	31/162 (19.7)	0.57 (0.29–1.10)
Sex												
Male	44/159 (27.7)	1	29/159 (18.2)	1	18/159 (11.3)	1	80/159 (50.3)	1	47/159 (29.6)	1	36/159 (22.6)	1
Female	231/545 (42.4)	2.15 * (1.44–3.21	178/545 (32.7)	2.22 * (1.42–3.45)	95/545 (17.4)	1.75 * (1.02–3.01)	267/545 (49.0)	0.96 (0.67–1.37)	167/545 (30.6)	1.05 (0.71–1.55)	137/545 (25.1)	1.20 (0.78–1.83)
Corneal parameters												
TBUT (≤5 s)	208/275 (75.6)	1.44 * (1.01–2.06)	158/207 (76.3)	1.36 (0.93–1.99)	84/113 (74.3)	1.19 (0.75–1.90)	251/347 (72.3)	1.16 (0.83–1.61)	177/214 (82.7)	2.53 * (1.69–3.80)	142/173 (82.0)	2.28 * (1.48–3.53)
Keratopathy score (≥3)	128/275 (46.5)	1.62 * (1.17–2.24)	102/207 (49.2)	1.74 * (1.25–2.43)	57/113 (50.4)	1.70 * (1.13–2.56)	132/347 (38.0)	0.93 (0.69–1.27)	82/214 (38.3)	0.99 (0.71–1.38)	68/173 (39.3)	1.00 (0.70–1.42)
Schirmer’s test value (≤5 mm)	94/275 (34.1)	1.39 (0.99–1.95)	71/207 (34.3)	1.22 (0.86–1.73)	28/113 (24.7)	0.71 (0.44–1.14)	100/347 (28.8)	1.02 (0.71–1.46)	69/214 (32.2)	1.08 (0.76–1.531)	42/173 (24.2)	0.66 (0.44–0.98)

Unless indicated otherwise, data are given as adjusted odds ratios (OR; adjusted for age and sex) with 95% confidence intervals (CI). * *p* < 0.05. TBUT, tear break-up time.

**Table 3 diagnostics-10-00193-t003:** Association between demographic characteristics and corneal signs.

	Short TBUT		Keratopathy Score		Low Schirmer’s Test Value	
	Prevalence (%)	OR (95% CI)	Prevalence (%)	OR (95% CI)	Prevalence (%)	OR (95% CI)
Age (years)						
≤29	40/67 (59.7)	1	33/67 (49.2)	1	13/67 (19.4)	1
30–39	51/82 (62.2)	1.15 (0.59–2.23)	28/82 (35.0)	0.58 (0.29–1.13)	19/82 (23.1)	1.25 (0.56–2.77)
40–49	84/107 (78.5)	2.63 * (1.33–5.20)	44/107 (41.5)	0.77 (0.41–1.44)	33/107 (30.8)	1.84 (0.88–3.84)
50–59	109/147 (74.1)	1.98 * (1.07–3.66)	49/147 (33.3)	0.52 * (0.28–0.95)	49/147 (33.3)	2.03 * (1.01–4.08)
60–69	107/139 (77.0)	2.24 * (1.18–4.24)	51/139 (36.6)	0.59 (0.32–1.08)	56/139 (40.3)	2.74 * (1.36–5.50)
≥70	108/162 (66.6)	1.34 (0.73–2.43)	66/162 (41.0)	0.71 (0.40–1.28)	46/162 (28.4)	1.60 (0.80–3.22)
Sex						
Male	93/159 (58.5)	1	45/159 (28.3)	1	42/159 (26.4)	1
Female	406/545 (74.5)	2.06 * (1.42–2.98)	226/545 (41.5)	1.82 * (1.23–2.68)	174/545 (31.9)	1.27 (0.85–1.89)

Unless indicated otherwise, data are given as adjusted odds ratios (OR; adjusted for age and sex) with 95% confidence intervals (CI). * *p* < 0.05. TBUT, tear break-up time.

**Table 4 diagnostics-10-00193-t004:** Summary of the association between age and dry eye-related signs and symptoms.

Age Group	Symptoms		Signs		
	Non-Visual	Visual	Keratopathy	Short TBUT	Aqueous Tear Deficiency
Younger (≤29 years)	+++	+	++	+	+
Middle-aged (30–49 years)	++	+	+	++	++
Older (≥50 years)	+	+	+	++	+++

The “+” symbols represent relative values compared with the other age groups. Eye fatigue, blurring, and photophobia were classified as visual symptoms; dryness, irritation, and pain were classified as non-visual symptoms. TBUT, tear break-up time.

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
