# Peer review of "Age Is a Determining Factor of Dry Eye-Related Signs and Symptoms"

_diagnostics, 2020, doi:10.3390/diagnostics10040193_

Round 1

Reviewer 1 Report

This study investigated the dry eye (DE) related symptoms and signs from different ages of patients. Total 704 consecutive patients were enrolled. The authors found that younger age was associated with non-visual symptoms and keratopathy. Older age was associated with short tear break-up time (TBUT) and low values on Schirmer’s test. Middle age was associated with both severe symptoms and signs. The authors concluded that discrepancies in the signs and symptoms of DE may depend on age partly.

  1. Please discuss the reason why different age of patients may present different signs and symptoms.

  1. Based on the results of this study, please discuss the different treatment may apply on different ages of patients.

  1. Were these participants been treated during the study?

Author Response

Thank you very much for your interest in our manuscript entitled “Age is a determining factor of dry eye-related signs and symptoms”. To aid in the re-review of this manuscript, we have included a point-by-point response to each comment. The reviewer’s comments are italicized and placed in square brackets. In addition, within the revised manuscript, we have used underlined text to highlight changes in response to the reviewers’ comments.

 [Reviewer 1 comment]

This study investigated the dry eye (DE) related symptoms and signs from different ages of patients. Total 704 consecutive patients were enrolled. The authors found that younger age was associated with non-visual symptoms and keratopathy. Older age was associated with short tear break-up time (TBUT) and low values on Schirmer’s test. Middle age was associated with both severe symptoms and signs. The authors concluded that discrepancies in the signs and symptoms of DE may depend on age partly.

We appreciate the reviewer’s complimentary comments.

 [Reviewer 1 Point 1: Please discuss the reason why different age of patients may present different signs and symptoms.]

We appreciate the reviewer’s complimentary comments. We discussed this issue in line147-159 in original submission and enhanced it by describing suggested treatment strategy based on the present results as follows.

[Discussion, page 5]

“Based on the present results, the different treatment may apply on different ages of patients. Anti-inflammatory eyedrops may be considered for younger patients suffering pain, irritation, and dryness. Mucin secretagogue should be first considered for older patients with tear dysfunction. Many middle-aged patients may be complicated with tear dysfunction and bothersome symptoms and mucin secretagogue and steroid may be prescribed for clinically significant signs and symptoms.”

 [Reviewer 1 Point 2: Based on the results of this study, please discuss the different treatment may apply on different ages of patients.

We appreciate the reviewer’s complimentary comments. We have responded to this comment as above.

 [Reviewer 1 Point 3: Were these participants been treated during the study?]

We appreciate the reviewer’s complimentary comment. They were treated and we have addressed the issues as follows.

[Results, page 3]

“The prescribed eyedrops for the treatment of DE were hyaluronate (0.1% sodium hyaluronate), mucin secretagogue (3% diquafosol sodium and 2% rebamipide), and steroid (0.02%/0.1% fluorometholon). Mucin secretagogue was significantly more prescribed for older groups than younger groups (P=0.041, Mann-Whitney U Test).”

Table 1. Age and sex distribution and dry eye medication of study participants.

Age (years)

Sex

Dry eye medication

Men (n = 159)

Women (n = 545)

Hyaluronate

Mucin secretagogue

Steroid

≤29

16 (10.1)

51 (9.4)

22 (32.8)

14 (20.9)

9 (13.4)

30–39

24 (15.1)

58 (10.6)

16 (19.5)

23 (28.0)

5 (6.1)

40–49

31 (19.5)

76 (13.9)

30 (28.0)

24 (22.4)

9 (8.4)

50–59

33 (20.8)

114 (20.9)

44 (29.9)

49 (33.3)

13 (8.8)

60–69

25 (15.7)

114 (20.9)

39 (28.1)

49 (35.3)

13 (9.4)

≥70

30 (18.8)

132 (24.3)

47 (29.0)

51 (31.5)

12 (7.4)

P value*

0.019

0.731

0.041

0.543

Data are presented as n (%). * Mann-Whitney U Test

Reviewer 2 Report

This study provide some information for ophthalmologist in daily clinic. The discrepancies in the signs and symptoms of DE may depend age, with severe non-visual symptoms with apparently normal tear function and severe keratopathy seen in younger subjects, and fewer symptoms and less severe keratopathy despite worse tear function in older subjects.

Author Response

Response to Reviewer 2

Thank you very much for your interest in our manuscript entitled “Age is a determining factor of dry eye-related signs and symptoms”. The reviewer’s comments are italicized and placed in square bracket.

 [Reviewer 2 Comments and Suggestions for Authors]

This study provide some information for ophthalmologist in daily clinic. The discrepancies in the signs and symptoms of DE may depend age, with severe non-visual symptoms with apparently normal tear function and severe keratopathy seen in younger subjects, and fewer symptoms and less severe keratopathy despite worse tear function in older subjects.

We appreciate the reviewer’s complimentary comments.